Grafting-enhanced tolerance of cucumber to toxic stress is associated with regulation of phenolic and other aromatic acids metabolism

Xiao Xuemei 1 2
Li Ju 2
Lyu Jian 2
Hu Linli 2
Wu Yue 2
Tang Zhongqi 2
Yu Jihua yujihua@gsau.edu.cn yujihuagg@163.com 1 2
Calderón-Urrea Alejandro 3 4
1 State Key Laboratory of Aridland Crop Science, Gansu Agricultural University , Lanzhou , China
2 College of Horticulture, Gansu Agricultural University , Lanzhou , China
3 College of Plant Protection, Gansu Agricultural University , Lanzhou , China
4 Department of Biology, College of Science and Mathematics, California State University , Fresno , CA, USA
Płachno Bartosz
Electronic publication date: 2022 Jun 1
Publication date: 2022
Volume: 10
Electronic Location ID: e13521
Received 2021 Nov 29; Accepted 2022 May 9
Copyright: ©2022 Xiao et al.
Copyright year: 2022
Copyright holder: Xiao et al.
License: This is an open access article distributed under the terms of the Creative Commons Attribution License, which permits unrestricted use, distribution, reproduction and adaptation in any medium and for any purpose provided that it is properly attributed. For attribution, the original author(s), title, publication source (PeerJ) and either DOI or URL of the article must be cited.
License URL: https://creativecommons.org/licenses/by/4.0/

Keywords: Cucumber, Grafting, Cinnamic acid, Phenolic and other aromatic acids biosynthesis

Funding: the Research Program Sponsored by State Key Laboratory of Arid-land Crop Science, Gansu Agricultural University GSCS-2020-12 National Natural Science Foundation of China 32160703 Natural Science Foundation of Gansu Province 20JR10RA513 General Project of Scientific Research in Colleges and Universities in Gansu Province 2021B-141 This work was supported by the Research Program Sponsored by State Key Laboratory of Arid-land Crop Science, Gansu Agricultural University (GSCS-2020-12), the National Natural Science Foundation of China (No. 32160703), Natural Science Foundation of Gansu Province (20JR10RA513), General Project of Scientific Research in Colleges and Universities in Gansu Province (2021B-141). The funders had no role in study design, data collection and analysis, decision to publish, or preparation of the manuscript.

==============================
Toxic stress caused by autotoxins is a common phenomenon for cucumber under monoculture condition. A previous study demonstrated that grafting could enhance the resistance of cucumber to cinnamic acid (CA) stress, but the underlying mechanism behind this enhanced resistance is still unclear. In the present study, we reconfirmed the stronger resistance of grafted rootstock (RG) compared to the non-grafted (NG) cucumber as measured though plant biomass accumulation. In addition, we focused on the phenolic and other aromatic acids metabolism in hydroponic culture model system using a combination of qRT-PCR (to measure gene expression of relevant genes) and HPLC (to detect the presence of phenolic and other aromatic acids). The results showed that the exogenous CA lead to the expression of four enzymes involved in phenolic and other aromatic acids biosynthesis, and a larger increase was observed in grafted rootstock (RG). Specifically, expression of six genes, involved in phenolic and other aromatic acids biosynthesis (PAL, PAL1, C4H, 4CL1, 4CL2 and COMT), with the exception of 4CL2, were significantly up-regulated in RG but down-regulated in NG when exposed to CA. Furthermore, six kinds of phenolic and other aromatic acids were detected in leaves and roots of NG and RG cucumber, while only benzoic acid and cinnamic acid were detected in root exudate of all samples. The CA treatment resulted in an increase of p-hydroxybenzonic acid, benzoic acid and cinnamic acid contents in RG cucumber, but decrease of p-coumaric acid and sinapic acid contents in NG cucumber. Surprisingly, the type and amount of phenolic and other aromatic acids in root exudate was improved by exogenous CA, particularly for RG cucumber. These results suggest that a possible mechanism for the stronger resistance to CA of RG than NG cucumber could involve the up-regulation of key genes involved in phenolic and other aromatic acids metabolism, and that the excessive phenolic compounds released to surroundings is a result of the accumulation of phenolic compounds in a short time by the plant under stress.

Introduction

Consecutive monoculture problem (CMP) is evident after repeatedly cultivating the same land under normal management practices, which results in a crop that can’t growth normally and produces less. CMP normally results in decline of plant growth and photosynthesis, and an increase of soil-borne diseases (Ye et al., 2006; Yu et al., 2003; Zhao et al., 2015). Autotoxicity is growth inhibition of plants caused by individuals of the same species, and has been demonstrated as a main cause of CMP (Singh, Batish & Kohli, 1999). The widely cultivated vegetable cucumber (Cucumis sativus L.) is commonly cultivated under greenhouse conditions, and suffers from serious consecutive monoculture problem. Ten phenolic and other aromatic acids were identified and isolated from cucumber root exudates as autotoxins, including cinnamic acid (CA), benzoic acid (BA), p-hydroxybenzoic acid (p-HBA) (Yu & Matsui, 1994). Numerous indoor and outdoor trials have demonstrated that single or multiple autotoxic compounds triggered oxidative stress (Ye et al., 2006), inhibition of photosynthesis (Yu et al., 2003), and alteration of rhizosphere microbial communities (Zhou & Wu, 2018; Zhou et al., 2018). However, the production and release of phenolic compounds induced by exogenous autotoxic compound remains unclear.

Phenolic acids consist of two major groups, hydroxybenzoic acids derivatives and hydroxycinnamic acids derivatives (Heleno et al., 2015). Production of phenolic and other aromatic acids starts from phenylalanine or tyrosine and goes through the three main reactions of deamination, hydroxylation and methylation in shikimic acid pathway (Deng & Lu, 2017; Shi et al., 2019). First, the deamination catalyzed by phenylalanine ammonia-lyase (PAL) and tyrosine ammonia-lyase (TAL) occurs from phenylalanine or tyrosine to cinnamic or p-coumaric acids. Afterwards, ferulic and caffeic acids are formed via hydroxylation and methylation of cinnamic and p-coumaric acids aromatic rings, and the same hydroxylation and methylation reactions happen in the benzoic acid to produce correspondent derivatives (Heleno et al., 2015). Several additional enzymes play a crucial role in the biosynthesis process of phenolic and other aromatic acids, such as cinnamic acid 4-hydroxylase (C4H), 4-coumarate–CoA ligase (4CL), benzoic acid 4-hydroxylase (B4H) and caffeic acid 3-O-methyltransferase (COMT) (Lin et al., 2020). The composition and content of phenolic and other aromatic acids can be affected by various factors, including developmental stage (Ma et al., 2016; Payyavula et al., 2013), stress condition (Şirin & Aslım, 2019), light (Qian et al., 2019) and fertilization (Jiménez-Gómez et al., 2020). Although, more and more studies focus on the antioxidant activity of phenolic and other aromatic acids due to their ability to scavenge reactive oxygen species (ROS) (Rice-Evans, Miller & Paganga, 1996; Hayat et al., 2009), the opposite seems to be the case since excessive accumulation of phenolic and other aromatic acids induces oxidative stress and autotoxicity, especially in susceptible plants (Blum, 1996). Therefore, it is of vital importance to understand the regulatory mechanisms involved in the production and release of phenolic and other aromatic acids.

Grafting is a traditional agronomic strategy used to enhance resistance to stress. Niu et al. (2019) demonstrated that grafting cucumber onto pumpkin improved osmotic tolerance under NaCl stress by upregulating ABA biosynthesis to induce early stomatal closure. Spanò et al. (2020) reported that grafted tomato showed stronger tolerance to an airborne virus infection than non-grafted tomato. Several studies confirmed grafting could help watermelon (Ling et al., 2013) and eggplant (Chen et al., 2011) to resist toxic stress caused by root exudate or autotoxic substances. Likewise, our previous work demonstrated that grafting with figleaf gourd provided a stable plant growth and photosynthetic activity of cucumber under cinnamic acid stress, and revealed the underlying molecular mechanism by transcriptome analysis (Xiao et al., 2020). On the basis of the transcriptomic data, we found that the phenylpropanoid biosynthesis was one of the top five pathways enriched as measured by differentially expressed genes (DEGs) under cinnamic acid treatment. Therefore, we hypothesized that grafting activate tolerance of cucumber to toxic stress by regulating the phenylpropanoid biosynthesis and altering the generation of phenolic and other aromatic acids.

In the present study, the four key enzymes involved in the biosynthesis of phenolic and other aromatic acids (PAL, C4H, 4CL and COMT) were measured and the expression of their coding genes were analyzed, and the composition and content of phenolic and other aromatic acids were measured in leaf, root and root exudates of grafted rootstock (RG) and non-grafted (NG) cucumber by HPLC. The results shed light on the graft-induced tolerance to autotoxins of cucumber as it relates to phenolic and other aromatic acids metabolism.

Materials and Methods

Plant materials and treatments

The experiment was conducted in the growth chamber with a day/night temperature of 28/21 °C, photoperiod of 14/10 h, relative humidity of 75%, and PPFD of 430 µmol m−2 s−1 with a LED light source. Cucumber (Cucumis sativus cv. ‘Xinchun No.4’) was grafted onto figleaf gourd (Cucurbia ficifolia Bouché), with a ’top insertion grafting’ method (Davis et al., 2008) (abbreviated as RG), whereafter non-grafted cucumber plants (abbreviated as NG) were planted 3 days later than the cucumber used as scion. When graft union completely healed seven days after grafting, the RG and NG cucumber were transferred to 1 L plastic container containing 1/2 Yamazaki nutrient solution (Yamazaki, 1978). The nutrient solutions were renewed at an interval of three days and aerated continuously by an air pump. At the three true-leaf stage, the RG and NG cucumber seedlings were treated with 0.5 mM cinnamic acid (CA), and no CA was added as a control (CK). The four treatment combinations were replicated three times with six plants in each replicate. Nine plants were randomly selected from each treatment to measure plant biomass seven days later. The third true leaf samples were collected at 12 h after exposure to CA treatment, frozen immediately in liquid nitrogen and stored at −80 °C until enzyme quantity and gene expression analysis.

Plant biomass

The samples were divided into shoot and root parts, washed with distilled water thoroughly, dried with tissue paper gently and weighed as fresh weight. Then they were undergone the deactivation of enzymes at 105 °C for 15 min and oven dried at 75 °C till a constant weight. At last, the dry weight was recorded.

Measurement of enzyme presence

Activities of six enzymes, including L-phenylalanine ammonia-lyase (PAL), cinnamate 4-hydroxylase (C4H), 4-coumarate: coenzyme A ligase (4CL) and caffeic acid O-methyltransferase (COMT) within the phenylpropanoid pathway were performed by sandwich ELISA according to the manufacturer’s instructions (Shanghai Enzyme-linked Biotechnology Co., Ltd.). Protein contents were measured according to the method of Bradford (Bradford, 1976). Three biological replicates were performed.

Quantitative real-time PCR (qRT-PCR)

The six genes participating in phenylpropanoid pathway, including PAL, PAL1, C4H, 4CL1, 4CL2, COMT, were selected based on the previous mRNA-Seq database to detect the transcript level. Total RNA was extracted with MiniBEST Plant RNA Extraction Kit (TaKaRa, China) following the manufacturer’s protocol. The first strand cDNA was synthesized using a two-step PrimeScript™ RT Master Mix (TaKaRa, China). The primer sequences used for qRT-PCR were listed in Table 1. The qRT-PCR was performed on LightCycler® 96 System (Roche, Switzerland) with SYBR® Premix Ex Taq™ II (TaKaRa, China) according to manufacturer’s instructions. The relative expression level was expressed as 2−ΔΔCT based on normalized against actin-7 (the internal standard gene) (Livak & Schmittgen, 2001).

Table 1 Sequences of the primers used for qRT-PCR.

Genes	Accession No.	Forward primer (5′ to 3′)	Reverse primer (3′ to 5′)	
PAL	XM_011659349.2	CGAGAGCAGCCATGTTGGTGAG	CCACGAAGAGGCAAGCAAGGAG	
PAL1	XM_004143207.3	CGGCAGCGGAGTCGTTGAAG	TCCGACAGCTCCACCACCAC	
C4H	NM_001305778.1	CCTCTCCGTCGTCCTCGCTATC	GACTTGGAGCCAGTTGCCGAAG	
4CL1	XM_004137598.3	CAGTTCCTTGCCTCGTAACACTCC	CCGCCGTTGCCTCTGGATTG	
4CL2	XM_004151860.3	GCCGTTGCTCTGCCGTTCTC	GAGCCACACTCGACACCATGTTC	
COMT	XM_004137156.3	GGCTCCGTCCAGAGGCTCTAC	CGAAGGCACTCATTCCGTAGGC	
actin-7	XM_004147305.3	GCCAGTGGTCGTACAACAGGTATC	AGCAAGGTCGAGACGGAGGATAG	

Determination of phenolic and other aromatic acids

Standard solutions and reagents

Phenolic and other aromatic acids of cinnamic acid (CA), p-coumaric acid (p-CA), p-hydroxybenzonic acid (p-HBA), benzoic acid (BA), ferulic acid (FA), caffiec acid (Caf A), gallic acid (GA) and sinapic acid (SA) were obtained from Sigma–Aldrich Inc. (St Luis MO, USA). Standard solutions of eight phenolic and other aromatic acids were prepared at a concentration of 1.0 mg ml−1 in methanol. HPLC-grade methanol and acetic acid were purchased from Fisher (USA). Water was purified by using a Milli-Q system (Millipore, Bedford, USA). Other chemicals and reagents used were of analytical grade.

Extraction and isolation of phenolic and other aromatic acids

Another eighteen plants for each treatment (RG-CK, RG-CA, NG-CK and NG-CA) was selected for collecting root extraction. Every three plants were placed in a 1 L plastic container as one replicate, and six replicates for one treatment. The residual nutrient solutions of the hydroponic cultures were collected at 36 h after CA treatment, respectively. At the end of exposure to CA, the RG and NG cucumber seedlings were taken out from the plastic container, and the third-true leaves and roots were collected, gently washed with distilled water, and dried with tissue for the determination of phenolic and other aromatic acids.

The phenolic and other aromatic acids of root exudates were extracted and isolated by a modified method based on Yu & Matsui (1993, 1994). The extraction and isolation of phenolic and other aromatic acids in leaves and roots of RG and NG cucumber seedlings was done according to the method of Abu-Reidah et al. (2012) and made a minor improvement.

Chromatographic conditions

Separation of phenolic compounds was performed on a Waters ACQUITY Arc system (Waters, USA) with a quaternary solvent delivery system, an autosampler, a column temperature controller and a UV detector. This instrument was equipped with a Waters Symmetry C18 (5 µm, 4.6 mm × 250 mm) and column temperature was maintained at 25 °C. The mobile phase was methanol (solvent A) and water containing 0.5% acetic acid (solvent B) with a gradient elution. The gradient program was as follows: 0–10 min, 30% A plus 70% B; 10–16 min, 50% A plus 50% B; 16–20 min, 70% A plus 30% B; 20–21 min, 100% A plus 0% B; 21–22 min, 30% A plus 70% B, with a flow rate 1.0 ml min−1. The injected volume was 10 µl and UV detection was at 280 nm.

Quantified with external standards

The standards of the identified phenolic and other aromatic acids were dissolved in methanol to construct the calibration curves. Quantification on a mass basis was based on peak area. The standard response curve for each phenolic and other aromatic acid was obtained with a linear regression. The seven concentrations of each phenolic and other aromatic acid was made with correlation coefficients >0.99.

Statistical analysis

The results are presented as means ± standard error (SE) in the figures. The statistical significance of differences was based on the least significant difference test (LSD, P <0.05) for an analysis of variance (ANOVA) using SAS software (version 8.1). Statistical different analysis was conducted between control and cinnamic acid treatment for non-grafted cucumber (NG) and grafted rootstock cucumber (RG), respectively.

Results

Plant biomass of NG and RG cucumber seedlings response to CA treatment

As shown in Fig. 1, the accumulation of shoot and root biomass were marked restricted for NG cucumber when exposure to 0.5 mM CA, while keep a more stable state for RG cucumber. Compared to control, fresh and dry weight in shoot of NG under CA treatment exhibited a decrease of 17.4% and 14.0%, respectively. Similarly, the fresh and dry weight in root of NG was reduced by 14.3% and 16.6%. Ratio of fresh and dry matter both in shoot and root of NG showed a significant decrease after exposed to CA treatment. However, only a significant decrease of 9.6% was observed in fresh weight of root for RG cucumber under control than CA treatment. Furthermore, we can see that high accumulation of plant biomass in grafted rootstock than non-grafted cucumber in the same conditions.

Figure 1 Effects of cinnamic acid (CA) on the plant biomass of non-grafted (NG) and rootstock grafted (RG) cucumber seedlings.

Samples were collected at 7 days after exposed to 0 mM (white bars) and 0.5 mM CA (dark bars). Data are means ± standard error of n = 9. Statistical different analysis was conducted between control and cinnamic acid treatment based on the least significant difference test (LSD). Bars sharing the same letters are not significantly different at the 5% level. (A–C) Fresh weight, dry weight and ratio of fresh and dry weight in shoot; (D–F) fresh weight, dry weight and ratio of fresh and dry weight in root.

Phenolic and other aromatic acids biosynthesis-related enzyme levels changes in NG and RG cucumber seedlings response to CA treatment

The PAL, C4H and 4CL are key enzymes of general pathway in phenylpropanoid biosynthesis, closely associated to the formation of cinnamic acid, benzoic acid and derivatives. Amount of three enzymes in NG and RG cucumber seedlings showed different response to cinnamic acid (Figs. 2A–2C). More PAL and 4CL were observed in RG under 0.5 mM CA treatment compared to 0 mM CA, with an increase of 41.0% and 54.5%. However, no significant difference was detected in NG after exposure to CA. The C4H quantity was not affected by CA treatment both in NG and RG cucumber. The COMT enzyme catalyzes the reaction of methyltransfer from caffeic acid to ferulic acid. The CA treatment significantly improved the COMT accumulation by 30.3% in RG, but no difference in NG cucumber seedling (Fig. 2D).

Figure 2 Effects of cinnamic acid (CA) on the enzyme production involved to phenolic acid biosynthesis of non-grafted (NG) and grafted rootstock (RG) cucumber seedlings.

Samples were collected at 12 h after exposed to 0 mM (white bars) and 0.5 mM CA (dark bars). Data are means ± standard error of n = 3. Statistical different analysis was conducted between control and cinnamic acid treatment based on the least significant difference test (LSD). Bars sharing the same letters are not significantly different at the 5% level. (A) PAL, Phenylalanine Ammonia Lyase; (B) C4H, Cinnamate 4-Hydroxylase; (C) 4CL, 4-coumarate Coenzyme A Ligase; (D) COMT, Caffeic aicd-O-methyltransferase.

Expression of six genes participating in phenolic and other aromatic acids biosynthesis in NG and RG cucumber seedlings response to CA treatment

The expression level of six genes coding enzymes in phenolic and other aromatic acids biosynthesis was measured using qRT-PCR. As shown in Fig. 3, different kinds of genes in NG and RG cucumber was different response to CA treatment. Expression of PAL and C4H showed similar change in NG or RG cucumber seedlings compared to corresponding controls. Lower levels of expression were found for PAL and C4H in NG under CA treatment, and yet a slight decrease in RG (Figs. 3A and 3C). Down-regulation of PAL1, 4CL1 and COMT were exhibited in NG, with a reduction rate of 36.7%, 61.0% and 86.4%. However, nearly two-fold up-regulation of PAL1 and 4CL1, and 1.5-fold up-regulation of COMT in RG were obtained exposed to CA compared to control (Figs. 3B, 3D and 3F). Surprisingly, expression level of 4CL2 significantly declined both in NG and RG when exposure to CA stress. Furthermore, more dramatic down-regulation was detected in NG, a decrease by 98.9% than control (Fig. 3E).

Figure 3 Effects of cinnamic acid (CA) on the enzyme coding genes expression involved to phenolic acid biosynthesis of non-grafted (NG) and rootstock grafted (RG) cucumber seedlings.

Samples were collected at 12 h after exposed to 0 mM (white bars) and 0.5 mM CA (dark bars). Data are means ± standard error of n = 3. Statistical different analysis was conducted between control and cinnamic acid treatment based on the least significant difference test (LSD). Bars sharing the same letters are not significantly different at the 5% level. (A) PAL, (B) PAL1, (C) C4H, (D) 4CL1, (E) 4CL2 and (F) COMT.

Phenolic and other aromatic acids in leaves, roots and root exudates of RG and NG cucumber seedlings

Total phenolic and other aromatic acids content

Total phenolic and other aromatic acids content was obtained from the sum of every kind of phenolic and other aromatic acid. As shown in Fig. 4, total phenolic and other aromatic acids content in leaf and root of non-grafted (NG) and rootstock grafted (RG) cucumber seedlings was accumulated after exposure to 0.5 mM CA. In particular for RG cucumber, a 1.41-fold and 4.08-fold increase was observed in leaf and root than control, respectively. Furthermore, we found that total phenolic and other aromatic acids content in leaf was extremely higher than that in root, nearly 20-fold.

Figure 4 Effects of cinnamic acid (CA) on the total phenolic acid content in leaf (A) and root (B) of non-grafted (NG) and rootstock grafted (RG) cucumber seedlings.

Samples were collected at 36 h after exposed to 0 mM (white bars) and 0.5 mM CA (dark bars). Data are means ± standard error of n = 3.

Composition of phenolic and other aromatic acids in leaves

Six kinds of phenolic and other aromatic acids in leaves of NG and RG cucumber seedlings exposed to 0 mM and 0.5 mM CA were determinated by HPLC, including gallic acid (GA), p-hydroxybenzonic acid (p-HBA), p-coumaric acid (p-CA), benzoic acid (BA), ferulic acid (FA) and cinnamic acid (CA) (Fig. 5). Different kinds of phenolic and other aromatic acids in NG and RG showed diverse response to CA treatment. No significant difference in GA, p-HBA, BA and FA was observed in NG cucumber between CA and CK. While the p-CA and CA contents in NG under CA treatment was 33.3% and 27.4% of lower than CK. For RG cucumber, p-HBA, BA and CA content significantly accumulated after exposed to CA, with an increase of 10-, 0.5- and 0.9-fold. The other three phenolic compounds exhibited no significant change.

Figure 5 Effects of cinnamic acid (CA) on the phenolic acids in leaves of non-grafted (NG) and rootstock grafted (RG) cucumber seedlings.

Samples were collected at 36 h after exposed to 0 mM (white bars) and 0.5 mM CA (dark bars). Data are means ± standard error of n = 3. Statistical different analysis was conducted between control and cinnamic acid treatment based on the least significant difference test (LSD). (A) Gallic acid, (B) p-hydroxybenzonic acid, (C) p-coumaric acid, (D) benzoic acid, (E) ferulic acid and (F) cinnamic acid.

Composition of phenolic and other aromatic acids in roots

The composition and content of phenolic and other aromatic acids in roots were different from that in leaves. Ferulic acid (FA) was not detected in RG cucumber, and gallic acid (GA), p-Hydroxybenzonic acid (p-HBA), p-coumaric acid (p-CA), sinapic acid (SA), and cinnamic acid (CA) were obtained in roots of both NG and RG cucumber (Fig. 6). For NG cucumber, only a higher FA content was observed under the treatment of CA compared to CK. Nevertheless, p-CA and SA contents were reduced by 26.7% and 39.1% when exposure to 0.5 mM CA. Meanwhile, the 38-fold of increase was found in CA content for RG cucumber under CA treatment. Surprisingly, the GA content declined by 32% in roots of RG after being exposed to CA, and FA was undetectable. The other three phenolic compounds were not affected by CA treatment.

Figure 6 Effects of cinnamic acid (CA) on the phenolic acids in roots of non-grafted (NG) and rootstock grafted (RG) cucumber seedlings.

Samples were collected at 36 h after exposed to 0 mM (white bars) and 0.5 mM CA (dark bars). Data are means ± standard error of n = 3. Statistical different analysis was conducted between control and cinnamic acid treatment based on the least significant difference test (LSD). (A) Gallic acid, (B) p-hydroxybenzonic acid, (C) p-coumaric acid, (D) benzoic acid, (E) sinapic acid and (F) cinnamic acid.

Composition of phenolic and other aromatic acids in root exudate

The root exudate was collected at 36 h after CA treatment in the hydroponic cultures. Result showed that only benzoic acid and cinnamic acid were detected in all samples (Table 2). No matter the kinds or the contents of phenolic and other aromatic acids were stimulated by exogenous CA for NG and RG cucumber. Compared to CK, two more phenolic compounds of p-Hydroxybenzonic acid (p-HBA) and p-Coumaric acid (p-CA) in NG, and four more phenolic compounds of p-HBA, p-CA, sinapic acid (SA) and caffeic acid (CafA) in RG were obtained after exposure to CA. The 28.0% and 351% of up-regulation of benzoic acid was observed in NG and RG when exposure to CA stress, respectively. The severe increase of cinnamic acid in root exudate was due to the addition of exogenous CA. In the treatment of 0.5 mM CA, equivalently, 74 mg CA was added into 1 L solution. As shown in Table 2, 49.340 mg L−1 and 84.546 mg L−1 of CA were detected in root exudate for NG and RG, indicating the absorption by NG and release of cinnamic acid by RG.

Table 2 Effect of cinnamic acid treatment on phenolic acids contents in root exudates (mg L −1) of RG and NG cucumber seedling.

Treatment	Gallic acid	p-Hydroxybenzonic acid	p-Coumaric acid	Sinapic acid	Benzoic acid	Caffeic acid	Ferulic acid	Cinnamic acid	Total phenolic acid	
NG-CK	0.024	–	–	–	0.318	–	–	0.455	0.797	
NG-CA	0.012	0.082	0.008	–	0.407	–	–	49.340	49.849	
RG-CK	–	–	–	–	0.280	–	–	1.438	1.718	
RG-CA	–	0.103	0.145	0.011	1.263	0.002	–	84.546	86.070	
Notes.

Root exudates for measuring phenolic acids contents were collected at 36 h after exposure to 0 mM and 0.5 mM cinnamic acid (CA).

NG-CK non-grafted cucumber seedling under control

NG-CA non-grafted cucumber seedling under CA stress

RG-CK rootstock grafted cucumber seedling under control

RG-CA rootstock grafted cucumber seedling under CA stress

Discussion

Consecutive monoculture problem happens commonly in cucumber cultivated in greenhouse. Previous studies demonstrated that autotoxicity is the main causes of consecutive monoculture problem (Yu & Matsui, 1994; Yu et al., 2003). Cinnamic acid, as the main autotoxin, was used to simulate autotoxic stress, and the inhibitory effect on growth was confirmed (Ye et al., 2006; Yu et al., 2003; Xiao et al., 2020). Although similar result were obtained in the present study, plant biomass of root-grafted cucumber seedlings was barely inhibited; similar findings were reported in eggplant (Chen et al., 2011) and watermelon (Ling et al., 2013). Given these observations, it is reasonable to explore whether grafting improves the resistance of cucumber to autotoxic stress by regulating metabolism of autotoxic substances.

Phenolic and other aromatic acids consists of aromatic acid compounds including the benzene ring linked with one or more hydroxyl or methoxy groups (Deng & Lu, 2017; Heleno et al., 2015). They are widely used in human daily diets due to its nutritional and high anti-oxidative capacity. Phenolic and other aromatic acids exhibit diverse biological functions, including plant–microbe symbiosis, plant allelopathic activities, resistance to pathogen attack and abiotic stress, and allow cross-linking with lignin to act as components of cell wall (Mandal, Chakraborty & Dey, 2010; Seal et al., 2004b). Comprehensive studies revealed that phenolic acids are potent autotoxins acted in a dose-dependent manner in many plants such as rice (Seal, Haig & Pratley, 2004a), soybean (Santos et al., 2008), alfalfa (Chon & Kim, 2002), watermelon (Hao et al., 2010) and cucumber (Yu & Matsui, 1994). It is well established that cucumber is sensitive to autotoxicity. Cinnamic acid, ferulic acid, p-hydroxybenzoic acid and p-coumaric acid were demonstrated as main autotoxic substances in cucumber, and they have an adverse effect on growth, physiology and soil microbial communities (Ye et al., 2006; Zhou & Wu, 2012; Zhou, Yu & Wu, 2012; Zhou & Wu, 2018). Our previous study also proved the detrimental effect of cinnamic acid on cucumber (Xiao et al., 2020). Biosynthesis of phenolic and other aromatic acids undergoes the general phenylpropanoid pathway (GPP) and subsequent specific branch pathways. A series of enzymatic reactions happen in the GPP process, and cinnamic acid, p-coumaric acid and p-coumaroyl-CoA are produced. The cinnamic acid is generated by deamination from phenylalanine under the action of phenylalanine ammonia lyase (PAL). In the study, the up-regulation of PAL production and coding genes expression and increase of cinnamic acid were observed in RG cucumber when exposure to CA (shown in Fig. 7). It was consistent with the result of Ye et al. (2006) and similar results were demonstrated after exposure to ferulic and p-coumaric acids (Politycka, 1999). Subsequently, cinnamic acid is hydroxylated to p-coumaric acid under the catalysis of cinnamate 4-hydroxylase (C4H). It is strange that the C4H quantity and gene expression showed no significant effect by CA in RG cucumber but the p-coumaric acids content increased (shown in Fig. 7). Therefore, we speculate that there is another pathway to produce p-coumaric acids, perhaps from tyrosine to p-coumaric acid under the catalysis of tyrosine ammonia lyase (TAL) (Barros et al., 2016). Following this, the conversion of p-coumaric acid into p-coumaroyl-CoA is catalyzed by 4-coumaroyl CoA ligase (4CL), which is an important branch point leading to generate phenolic compounds (Deng & Lu, 2017; Liu, Osbourn & Ma, 2015). For NG cucumber, CA significantly enhanced the 4CL production and coding gene expression, while the expression of 4CL genes was down-regulated in NG cucumber (shown in Fig. 7). This may be the explanation of why the decrease of p-coumaric acid content while relatively stable cinnamic acid and benzoic acid contents was found in NG exposed to CA. Actually, phenolic acid biosynthesis is less clear than flavonoids and monolignols, and many steps of the pathway are unknown. Hanson & Gregory (2011) has reported that benzoic acid is derived from cinnamic acid through multiple routes, in which two carbon units from the C3 side chain of cinnamic acid is degraded. Therefore, the amount of cinnamic acid can influence the production of benzoic acid and its derivative. Caffeic acid O-methyltransferase (COMT) catalyzes the conversion of caffeic acid into ferulic acid. Our study showed that caffeic acid was not detected in all samples and ferulic acid content was increased under 0.5 mM CA treatment than control for NG. However, the remarkable up-regulation of COMT production and coding gene expression in RG while the down-regulation of COMT expression were observed in NG. It suggests another unknown enzyme or pathway involved in biosynthesis and conversion of ferulic acid.

Figure 7 Schematic representation of phenolic acids metabolism in rootstock grafted cucumber (RG) and non-grafted cucumber (NG) response to CA stress.

Red represents up-regulation of gene expression and increase of phenolic acids content, while green as down-regulation and decrease.

Most of phenolic and other aromatic acids have been reported to be antioxidant agents against free radicals, so they play a vital role in plant resistance to biotic and abiotic stress (Heleno et al., 2015). Huang et al. (2019) described that levels of benzoic acid, gallic acid hydrate and p-coumaric acid were up-regulated to compensate the depletion of the other antioxidants in cucumber leaf tissues after exposed to nano copper pesticide. Similar results also were reported by Simek et al. (2016), where phenolic compounds had risen under cadmium stress in cucumber plants. Furthermore, the study of Chen et al. (2013) explored the potential of AMF in the alleviation of chilling stress for cucumber seedlings, and discovered AMF symbiosis could decrease hydrogen peroxide (H2O2) content by accumulation of secondary metabolites including phenolic compounds and up-regulation of related enzymatic activities and genes expression under low temperature. The present study showed the increased amounts of enzymes and coding gene expression involving in phenolic and other aromatic acids biosynthesis under CA treatment. It indicated exogenous cinnamic acid could stimulate the antioxidant system of cucumber to scavenge reactive oxygen species. The higher enzyme quantity, gene expression and content of phenolic and other aromatic acids compounds was obtained in RG than NG cucumber (shown in Fig. 7), which explained the stronger tolerance to autotoxic stress when cucumber was grafted onto figleaf gourd. It is consistent with the result of Rivero, Ruiz & Romero (2003) in tomato plants under thermal stress, and Sánchez-Rodríguez et al. (2011) in cherry tomatoes under water stress. An interesting discovery was obtained in our study that more phenolic and other aromatic acids was exuded from the root of RG than NG cucumber (shown in Fig. 7). We deduce that it is another reason why grafting improve the resistance of cucumber to autotoxic stress. However, the underlying mechanism of the difference in exudation and release of phenolic and other aromatic acids between RG and NG cucumber need further study.

Conclusions

The phenolic and other aromatic acids metabolism of non-grafted (NG) and grafted rootstock (RG) cucumber seedling presented different response to cinnamic acid (CA) stress. Figure 7 summarizes the change of key enzymes involving in phenolic and other aromatic acids metabolism, and the production and release of phenolic and other aromatic acidsin NG and RG cucumber seedlings when exposed to CA. We found that exogenous CA induced up-regulation of PAL, 4CL and COMT, and accumulation of cinnamic acid, benzonic acid and p-hydroxybenzoic acid in RG cucumber. The four key enzymes were all down-regulated in NG, and p-coumaric acid and sinapic acid content were decreased. Surprisingly, more ferulic acid was present in NG after exposure to CA. It suggested that another unknown pathway could produce the ferulic acid or transformation of ferulic acid were inhibited. Moreover, the type and amount of phenolic and other aromatic acids in root exudate was improved by exogenous CA especially for RG cucumber. In sum, the results suggest that the likely reason for stronger resistance to CA of RG cucumber could be firstly rapid production of phenolic and other aromatic acids, which then can be release to the surroundings.

Supplemental Information

Supplemental Information 1 Raw data including plant biomass, key enzymes activities, gene expression and substances involving in phenolic acid metabolism

Click here for additional data file.

We appreciate Xiaoqi Liu and Dan Zhang for their kind suggestions to perfect the method of phenolic and other aromatic acids.

Additional Information and Declarations

Competing Interests

Author Contributions

Data Availability

The authors declare there are no competing interests.

Xuemei Xiao conceived and designed the experiments, performed the experiments, analyzed the data, prepared figures and/or tables, authored or reviewed drafts of the article, and approved the final draft.

Ju Li performed the experiments, analyzed the data, prepared figures and/or tables, and approved the final draft.

Jian Lyu performed the experiments, authored or reviewed drafts of the article, and approved the final draft.

Linli Hu analyzed the data, authored or reviewed drafts of the article, and approved the final draft.

Yue Wu analyzed the data, prepared figures and/or tables, and approved the final draft.

Zhongqi Tang performed the experiments, prepared figures and/or tables, and approved the final draft.

Jihua Yu conceived and designed the experiments, authored or reviewed drafts of the article, and approved the final draft.

Alejandro Calderón-Urrea analyzed the data, authored or reviewed drafts of the article, and approved the final draft.

The following information was supplied regarding data availability:

The raw measurements are available in the Supplementary File.

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
