# Peer review of "Grafting-enhanced tolerance of cucumber to toxic stress is associated with regulation of phenolic and other aromatic acids metabolism"

_PeerJ, doi:10.7717/peerj.13521_

## Round 0.1 · original submission · Major Revisions

Dear Authors,
Please respond to all suggestions and changes proposed by reviewers. As the third reviewer has serious doubts about the design of the experiment, I am asking for a detailed explanation.

Best Regards,
Bartosz Płachno

Reviewer 1 ·

Basic reporting

In the manuscript entitled "Grafting enhances tolerance of cucumber to toxic stress by regulating phenolic acid metabolism" its authors focused on elucidating the phenolic acid metabolism, using the method of qRT-PCR and HPLC, using Cucumis sativus cv. ‘Xinchun No.4’ which was treated with 0.5 mM of cinnamic acid (CA) in the hydroponic culture. The manuscript submitted to Peer J fits to the scope of journal, but need to be checked by a proficient English speaker because there are sentences that need to be reworded, especially in Abstract, Discussion and Conclusions. Figures are relevant to the contents of the manuscript. Figure 1 needs improvement. here, the ratio of fresh and dry matter should be given. The manuscript include sufficient Introduction and relevant literature.

Experimental design

The experimental design is adequate to this type of work. The phenolic acid metabolism of non-grafted (NG) cucumber seedlings and those grafted (RG) onto Cucurbia ficifolia presented different response to added cinnamic acid. Authors have ascertained that exogenous supplementation of CA induced up-regulation of PAL, 4CL and COMT, and accumulation of cinnamic acid, benzonic acid and p-Hydroxybenzoic acid in RG cucumber plants. In the case of NG the four key enzymes were all down-regulated , p-coumaric acid and sinapic acid content were decreased, but still the level of ferulic acid was relatively high. T

Validity of the findings

The authors have suggested that either an unknown pathway could produce the ferulic acid or transformation of ferulic acid were inhibited. Moreover, the type and amount of phenolic acid in root exudates was improved by exogenous CA especially for RG cucumber. Their results point that the possible reason for possible resistance to CA of RG cucumber could be rapid production of phenolic acid or its dynamic release from roots system to the rhizosphere.

Additional comments

Some detailed comments:
Lines 25-26: ..."we focused on the phenolic acid metabolism using the method of qRT-PCR and HPLC in hydroponic culture"... REALLY! HPLC in hydroponic culture ?
line 178: ..." 16-20min"... INSERT SPACE
line 263: ... up- regulation" REMOVE SPACE
Dissussion: Not all results have been discussed!
line 269: ..."Phenolic acids are a common group of phenylpropanoids, consist of aromatic acid compounds"... THAT CONSIST
line 271-275: The begging of the sentence should be: They are used distributed in human diet....
...and have been verified the activities of multiple nutrition and pharmacology, particularly 273 high anti-oxidant capacity - this is hard to understand what do you mean - please, reword this sentence!
Next sentence should probably be:: Phenolic acids exhibit diverse biological functions, including plant-microbe symbiosis, plant allelophatic activity, resistance to pathogen attack and abiotic stress, and allow cross-linking with lignin to act as components of cell wall.
And so on and the like
1ine: 290: were demonstrated
line 315: plant resistance/tolerance
line 219: ..."Similar results also were reported by Simek et al. (Simek et al., 2016), phenolic compounds rose under cadmium stress in cucumber plants".... HAD ROSEN
line 333: ..." We deduce that it is another reason for grafting improve the resistance"... - Please write the sentence correctly

Reviewer 2 ·

Basic reporting

The authors investigate the effect of grafting on tolerance to toxic stress (autotoxins) in cucumber monoculture through the regulation of phenolic acid metabolism.This study builds on a previous study, including transcriptomics data, which sheds light on this graft-induced tolerance. The writing is clear, unambiguous and technically correct. The structure of the article has an acceptable format for PeerJ.

The introduction has sufficient and up-to-date background information. Only one thing needs to be clarified: "We found more phenolic acid..." (lines 91-93). I think this sentence does not belong in the introduction section.

The figures and tables chosen by the authors are adequate to show the results of the research. The graphs are of sufficient resolution (although the font size of the axis titles should be increased) and are correctly described and labelled. However, in all figure captions the authors did not indicate what each letter is (A, B, C, D). Although I can understand the figures with the axis titles, this should be corrected by including the meaning of the letters in the figure captions.

Experimental design

Manuscript clearly shows the research question in lines 85-87. Methods are well described. Experimental design is correct. I have some recommendations about Material and Methods section:

o I recommend the use of PPFD units (μmol/m2/s) and not lux (illuminance) units (line 98).
o When is the "graft union completely healed" (line 101)? How many days are necessary for this after grafting?
o Are the "shoot and root parts" (line 112) equivalent to the scion and rootstock parts in the case of grafted plants?
o Which part of the plant (first true leaf, second true leaf, whole shoot...) was selected for the enzyme activity assays?

Validity of the findings

The results are well described and correspond to the figures. The authors should correct the CA concentration (0.25 should be 0.50 mM) in lines 195 and 208. The discussion is of sufficient quality. The conclusions are adequate, connected to the original research question and supported by the results. However, I suggest that the authors send figure 7 to the discussion section.

Additional comments

No additional comments.

Reviewer 3 ·

Basic reporting

The work of Xiao et al. deals with the potentially interesting topic of cucumber cultivation.
However, the work has a series of flawns that depreciate the value and lower the quality of the manuscript.
1. The language quality is low. The text lacks clarity; it's challenging to follow the author's way of thinking.
2. The references need to be tidied out. There are editorial problems. The background of the problem is not sufficiently presented.
3. There is a lack of sufficient justification of the novelty of work and importance of problem presented (ma by caused by the low quality of language)
4. There is an essential issue of phenolic acids. Please be informed that benzoic acid and cinnamic acids are not phenolic acids. Compound to be classified as the phenolic compound has to have.

Experimental design

There are severe allegations of to method used.
1. Plant vegetation conditions are insufficiently described. Some remarks:
- lighting environment - what was a light source, light intensity reported as illuminance is not appropriate
Please be warned that lighting intensity has a significant impact on phenolics metabolism.
2. How to measure enzyme activity by sandwich ELISA???
3. Extraction of phenolic and benzoic acids from exudates has no sense. Liquid-liquid extraction (LLE) from acidic water solution followed by LLE from basic water solution is odd. What was the purpose?
Why is the extraction from plant material done another way?

Validity of the findings

Based on previous remarks findings cannot be validated.

---

## Round 0.2 · Minor Revisions

Dear Authors,

Because, both reviewers think that your work was substantially improved, I am pleased to inform you that your paper will be accepted after minor revision.

Best Wishes,
Bartosz Płachno

Reviewer 2 ·

Basic reporting

The authors have replied satisfactorily to my comments.

Experimental design

The authors have replied satisfactorily to my comments.

Validity of the findings

The authors have replied satisfactorily to my comments.

Additional comments

None

Reviewer 3 ·

Basic reporting

The work was substantially improved.
The only obstacle to publication is for me that authors consequently use the terms "phenolic acids" or "phenolic compounds" regarding cinnamic acid and benzoic acid. Those are not phenolics, as they do not directly have a hydroxyl group attached to the aromatic ring. Generally, all compounds authors denote as phenolics are aromatic acids.
The use of "phenolics" regarding virtually any aromatic secondary metabolite seems to be a common mistake in the literature and promotes incorrect biochemical/physiological reasoning.
My compromise proposal is the term: "phenolic and other aromatic acids" or "phenolic and other related aromatic acids".

Experimental design

The work was substantially improved.

Validity of the findings

The work was substantially improved.

---

## Round 0.3 · Minor Revisions

Dear Authors, thank you for correcting the article as directed by the reviewer. Congratulations, the article has been accepted after one small correction.

Please change the title for something like: 'Grafting-enhanced tolerance of cucumber to toxic stress might be due to regulation of phenolic and other aromatic acids metabolism' or 'Grafting-enhanced tolerance of cucumber to toxic stress is associated with regulation of phenolic and other aromatic acids metabolism'. This would more accurately reflect the results of the paper.

Kind regards,
Bartosz Płachno

---

## Round 0.4 · accepted · Accept

Dear Authors,
Thank you very much for the correction. I am pleased to inform you that your ms has been Accepted for publication.

Best Wishes,
Bartosz Płachno